# Rhenium(I) Block Copolymers Based on Polyvinylpyrrolidone: A Successful Strategy to Water-Solubility and Biocompatibility

**DOI:** 10.3390/molecules28010348

**Published:** 2023-01-01

**Authors:** Kristina S. Kisel, Vadim A. Baigildin, Anastasia I. Solomatina, Alexey I. Gostev, Eugene V. Sivtsov, Julia R. Shakirova, Sergey P. Tunik

**Affiliations:** 1Institute of Chemistry, Saint-Petersburg State University, Universitetskii pr., 26, 198504 St. Petersburg, Russia; 2Department of Physical Chemistry, Saint-Petersburg State Institute of Technology, Technical University, Moskovskiy pr. 26, 190013 St. Petersburg, Russia

**Keywords:** rhenium complexes, polyvinylpyrrolidone, RAFT polymerization, phosphorescence, water-solubility, biocompatibility

## Abstract

A series of diphosphine Re(I) complexes **Re1**–**Re4** have been designed via decoration of the archetypal core {Re(CO)_2_(N^N)} through the installations of the phosphines **P^0^** and **P^1^** bearing the terminal double bond, where N^N = 2,2′-bipyridine (**N^N1**), 4,4′-di-tert-butyl-2,2′-bipyridine (**N^N2**) or 2,9-dimethyl-1,10-phenanthroline (**N^N3**) and **P^0^** = diphenylvinylphosphine, and **P^1^** = 4-(diphenylphosphino)styrene. These complexes were copolymerized with the corresponding N-vinylpyrrolidone-based Macro-RAFT agents of different polymer chain lengths to give water-soluble copolymers of low-molecular **p(VP-l-Re**) and high-molecular **p(VP-h-Re)** block-copolymers containing rhenium complexes. Compounds **Re1**–**Re4**, as well as the copolymers **p(VP-l-Re)** and **p(VP-h-Re)**, demonstrate phosphorescence from a ^3^MLCT excited state typical for this type of chromophores. The copolymers **p(VP-l-Re#)** and **p(VP-h-Re#)** display weak sensitivity to molecular oxygen in aqueous and buffered media, which becomes almost negligible in the model physiological media. In cell experiments with CHO-K1 cell line, **p(VP-l-Re2)** and **p(VP-h-Re2)** displayed significantly reduced toxicity compared to the initial **Re2** complex and internalized into cells presumably by endocytic pathways, being eventually accumulated in endosomes. The sensitivity of the copolymers to oxygen examined in CHO-K1 cells via phosphorescence lifetime imaging microscopy (PLIM) proved to be inessential.

## 1. Introduction

The growing interest in luminescent transition metal complexes stems from their potential application in multimodal bioimaging as versatile platforms since they may combine excellent spectroscopic characteristics and biologically relevant functionalization in a single molecular core [1,2]. The judicious choice of functional ligands and metal centers make it possible to prepare phosphorescent coordination compounds, which possess bright long-lived emissions with a large Stokes shift and low photobleaching that is of crucial importance for their application in luminescence imaging of various intracellular structures and tissues [3]. The combination of a long emission lifetime and sizeable Stokes shift enables better energy and lifetime separation of the label phosphorescence and autofluorescence of bio-samples that improves the quality of resulting images [4,5]. Additionally, the choice of the appropriate ligand environment is an effective instrument in fine tuning of physicochemical characteristics, such as water-solubility, stability in physiological media, low toxicity, and affinity to certain cell/tissue compartments [6].

In this respect, diimine rhenium(I) complexes are among the most fascinating candidates for luminescent materials suitable for biomedical applications, owing to their ability to fit the requirements listed above [5,7,8]. As a rule, the archetypal chromophores [Re(N^N)(CO)_3-x_L_x_] reveal the ^3^MLCT-originated photoluminescence, whose efficiency can be controlled by tuning the steric and/or electronic properties of the diimine N^N ligand [9]. Incorporation of the auxiliary ligands (L) with functional groups into the Re-based core is a typical way to expand the functionality of rhenium(I) compounds and to obtain the desired physicochemical characteristics. Solubility of the final products in aqueous/physiological media and biocompatibility are the most important characteristics, however, achieving this is still challenging.

The most obvious pathway to make the complexes soluble in aqueous/buffered solutions includes the incorporation of the hydrophilic phosphine ligands, such as Na_3_-tris-(3-sulfophenyl)phosphine, 1,3,5-triaza-7-phosphaadamantane, and etc., into tricarbonyl Re(I)-species [10,11,12]. However, this strategy does not always lead to a successful internalization of the target Re(I) compounds into cells. Additionally, such relatively low molecular Re(I) compounds may interact with biomolecules in physiological environments that may significantly change their photophysical properties.

An alternative approach to impart water-solubility, which avoids the aforementioned side effects, consists of the conjugation of rhenium chromophores with water-soluble polymers. Generally, the conjugation of metal moieties with polymer chain can be performed either by the direct participation of the ligand function in polymerization reaction [13,14,15,16] or by the complex bonding to side group functionalities of the polymer main chain [17]. Following this general concept, two alternative pathways to prepare metal-containing polymer architectures are feasible. The first approach utilizes the uncomplexed ligands to obtain polymers followed by complexation between the resulting macroligands and the metal units. The second option consists in the polymerization reaction via the function of one or more ligands in the coordination environment of the metal center. Among the transition metal containing copolymers, the rhenium-based polymer macrocomplexes remain scarce and are represented by a limited number of the Re(I) systems with polyvinylbipyridine as a macroligand [16,18]. However, modification of the diimine ligands is a rather complicated and demanding task and can lead to dramatic changes in the photophysical behavior of the target Re(I) compounds compared to the initial rhenium species due to involvement of the diimine fragment into a ^3^MLCT emissive excited state.

Therefore, herein we suggest an alternative pathway to obtain water-soluble Re-containing polymers based on a new synthetic strategy involving the preparation of the luminescent rhenium(I) complexes bearing the ancillary phosphine ligands with the alkenyl group followed by the copolymerization of the metal containing compounds with polyvinylpyrrolidone, p(VP), being used as a macro reversible-addition fragmentation transfer (RAFT) agent. Negligible toxicity and water solubility of p(VP) made it a widely used carrier to deliver hydrophobic therapeutic agents to targets [19]; this prompted us to use it in the construction of block-copolymers featuring desired physicochemical properties and biocompatibility. Direct integration of the Re(I) chromophore into the polymer chain leads to biocompatible emissive copolymers with increased stability due to strong resistance of covalent bonding to hydrolysis processes that is of considerable importance for application in biological experiments. All Re(I) species and obtained copolymers were fully characterized, and their photophysical properties were carefully investigated. The synthesized Re(I)-macromolecular block copolymers demonstrate a negligible response to variations in oxygen concentration in the model physiological media and keep nearly unchanged in their luminescence efficiency due to effective shielding of Re-based emitters from side interaction with physiological media components. Cellular experiments with Chinese hamster ovary (CHO-K1) cells demonstrated the suitability of the Re-containing block copolymers for bioimaging experiments. 

## 2. Results and Discussion

### 2.1. Synthesis and Characterization

#### 2.1.1. Complexes **Re1**–**Re4**

The synthesis of four new bis-phosphine Re(I) complexes required extreme temperature conditions. The reaction of the [Re(CO)_3_(N^N)(NCMe)]OTf precursors with the corresponding phosphine ligands was carried out in boiling o-dichlorobenzene in the presence of butylhydroxyanisole (BHA) to prevent the vinyl group thermal self-polymerization as shown in Figure 1. This approach enabled the synthesis of various Re(I)-based complexes as the parent compounds for preparation of the macromolecular metal complexes. The choice in favor of the titled diimines was dictated by high quantum efficiency of the related Re(I) species [20], whereas the phosphine ligands with the vinyl/styrene groups have been used as convenient participants of a copolymerization reaction between the rhenium phosphine complexes and polyvinylpyrrolidone, aimed at the solubilization of the target compounds in aqueous media. The yields of these reactions vary from moderate to good (up to ca. 93%) for all Re(I) phosphine derivatives.

The synthesized Re(I) complexes were characterized using a range of standard spectroscopic techniques, confirming the proposed formulations in each case. The relevant details and analytical/spectroscopic data are presented in the Experimental section and in the Electronic Supporting Information. IR spectra of the complexes in dichloromethane solution (Appendix A) reveal two bands, ca. 1870 and 1940 cm^−1^, typical for stretching vibrations of coordinated carbonyl ligands in the [Re(CO)_2_(N^N)L_2_]^+^ species, suggesting the local C_2ν_ symmetry of these molecules. The CO stretching frequencies display very small variations in this series of compounds, showing only slight changes in the donor ability of the diimine and phosphine ligands across the whole series of the rhenium complexes. The ^1^H and ^31^P{^1^H} NMR spectroscopic patterns indicate the presence of a symmetrical stereochemical environment around the Re(I) metal center in the final compounds (Appendix A). The ^31^P NMR spectra of **Re1**–**Re4** demonstrate singlet resonances that is completely compatible with trans-disposition of two equivalent phosphine ligands. The ^1^H NMR spectra indicate that there is no apparent steric hindrance, which could prevent free rotation around the P–Re–P axis. The distinguishable low-field resonances in the ^1^H NMR spectra correspond to the protons of the diimine ligand, whereas those detected in the range of 6.75–5.35 ppm are generated by the alkene protons of the phosphine ligands. The unresolved resonances in the aromatic region correspond to the protons of P-bonded phenyl rings. For the N^N ligands incorporating the alkyl substituents, the -C(CH_3_)_3_ and -CH_3_ resonances appear around 1.35 and 2.43 ppm, respectively, and were reasonably insensitive to coordination to Re(I). ESI^+^ mass spectrometry reveals the dominating signals that correspond to molecular ions of a general stoichiometry [Re(CO)_2_(N^N)P_2_]^+^ at *m*/*z* 823.17 for [**Re1**]^+^, 975.23 for [**Re2**]^+^, 1087.35 for [**Re3**]^+^, and 1027.26 for [**Re4**]^+^ with an experimental isotopic pattern that matches well the calculated values (Appendix A).

The symmetrical structure of complexes **Re1** and **Re2** was also confirmed by XRD analysis. Single crystals suitable for the X-ray crystallographic study were obtained by gas-phase diffusion of diethyl ether into a chloroform solution of these complexes. An Oak Ridge Thermal Ellipsoid Plot (ORTEP) view of complexes **Re1** and **Re2** is shown in Figure 1, and selected bond lengths and angles are summarized in Appendix A.

As expected, the structures of **Re1** and **Re2** display slightly distorted octahedral geometry of Re(I) centers, typical for this type of rhenium complexes [21,22,23]. In these structural patterns, the Re(I) ion is surrounded by the sterically demanding N^N chelating unit, trans-coordinated phosphine ligands, and two carbonyl groups, as shown in Figure 1. The interatomic bond lengths and angles around the Re(I) center are not exceptional and lie in the range found for the other bis-phosphine species (Appendix A). A noticeable structural feature of the synthesized Re(I) systems is a sterically accessible alkenyl group directly bonded either to the phosphorus atom or to the phenyl ring of the phosphine ligand in **Re1** and **Re2**, respectively. The C = C bond length in **Re1** and **Re2** (ca. 1.30 Å) is only slightly shorter than in the free ethylene (1.33 Å), that presumably indicates the retention of the electronic properties and, as a consequence, the reactivity of the terminal double bond in polymerization reactions. Therefore, we have examined the ability of all obtained complexes to take part in the formation of their copolymers with polyvinylpyrrolidone.

#### 2.1.2. Copolymers **p(VP-l-Re)**, **p(VP-h-Re)**

The direct copolymerization of the rhenium complexes containing the alkenyl functions to a hydrophilic polymer chain is a challenging task in obtaining the phosphorescence probes with the properties that make them suitable for application in bioimaging. However, this approach can reduce the amount of preparation stages and, in addition, ensures water-solubility and low toxicity of the rhenium complexes due to the use of biocompatible polymers.

Our choice was to use the reversible addition-fragmentation chain transfer (RAFT) block copolymerization of the rhenium complexes with polyvinylpyrrolidone p(VP), which was used as a macro-RAFT agent. p(VP) was selected due to its long-term stability, water-solubility, low toxicity [19], and reduced oxygen permeability [24,25]. The initial p(VP) and the corresponding block copolymers were prepared as two series with different lengths of the p(VP) chains based on application of two RAFT agents: 1-(*O*-ethylxanthyl)methylbenzene (RAFT1, low molecular polymer) [26] and *S,S*-dibenzyltrithiocarbonate (RAFT2, high molecular polymer) [27]. As a result, the metal containing block copolymers contain either one p(VP) fragment (low molecular block-copolymer, **p(VP-l-Re)**), or two p(VP) fragments (high molecular block-copolymer, **p(VP-h-Re)**), that allows the analysis of the influence of the p(VP) molecular weight on the photoluminescence behavior of the Re(I)-containing copolymers.

The synthetic routes and chemical structure of the obtained Re(I)-containing copolymers are shown in Figure 2. At the first stage, the low (**p(VP-l)**) and high (**p(VP-h)**) molecular polyvinylpyrrolidone macro-RAFT agents were prepared. Their mass average molecular weight was determined using gel-permeation chromatography (GPC, Table 1) to give 4600 and 67000 D for (**p(VP-l)**) and (**p(VP-h)**), respectively. The ^1^H NMR spectra of these products (Appendix A) are completely compatible with the structural patterns shown in Figure 2. Interestingly, **p(VP-l)** displayed a polydispersity coefficient (***Ɖ***) as low as 1.14 while the magnitude obtained for **p(VP-h)** was 1.59, which indicates better compatibility of RAFT1 to N-vinylpyrrolidone, in agreement with the results published earlier [28].

At the next stage, **Re1**–**Re4** were copolymerized with (**p(VP-l)**) and (**p(VP-h)**); in these reactions the polymeric precursors act as macro-RAFT agents to give the target block copolymers. It is worth noting that the first series presents the di-block-copolymers, whilst the second series forms the tri-block copolymers, where the Re-fragment is a central part (Figure 2).

Since the difference in the number of protons in the p(VP) blocks and in the Re complexes is very large, the signals from the emitter ligands in the ^1^H NMR spectra (Appendix A) are nondetectable. However, the presence of the signals in the ^31^P NMR spectra (Appendix A) clearly indicates the conjugation of the Re(I) compounds to the p(VP) copolymers. The positions of the ^31^P NMR signals are in agreement with those found for the initial (non-polymerized) metal complexes (Appendix A). The copolymerization of rhenium complexes has also been monitored by the disappearance of the double bond signal (5.88 ppm, **Re2**) in the ^1^H NMR spectra in the course of reaction. The time dependence of conversion (q) was calculated using Equation S1 and is shown in Figure 2. It was established that in the case of **Re2** copolymerization the conversion reaches saturation in 6 h to give the final magnitude of ca. 0.6. It is worth noting that due to the limitations of NMR measurements the monitoring was carried out in the less concentrated solutions, compared to the synthetic reaction conditions, which means that the conversion in the latter case is most probably to be even higher.

IR spectra of the polymeric metal complexes **p(VP-l-Re2)**–**p(VP-l-Re4)** and **p(VP-h-Re2)**–**p(VP-h-Re4)**, obtained using the Attenuated Total Reflection method, revealed the signal characteristics of CO stretching frequencies at 1860 cm^−1^ and 1935 cm^−1^ typical for the {Re(CO)_2_(N^N)P_2_} structural motifs (Appendix A). Low content of rhenium centers in the final copolymers **p(VP-l-Re1)** and **p(VP-h-Re1)** did not allow detecting the corresponding CO bands in the IR spectra.

Quantification of the metal loading in the copolymers was carried out via Ultraviolet–visible spectroscopy (UV-Vis), since the intensity of the characteristic absorption maxima increases with the increase in the ratio of the metal complex to the final products. The obtained data are in agreement with the values determined by Inductively Coupled Plasma Optical Emission Spectroscopy (ICP-OES) (Table 1). It should be noted that the data obtained cannot give the exact molar ratio of Re-block to p(VP)-block because of the peculiarities of the copolymer purification process. This procedure allows us to separate the unreacted rhenium complex, but not p(VP), and in the final calculations we cannot take into account the mass of unreacted p(VP). Nevertheless, the data show that **Re1** displays lower reactivity in the copolymerization reaction compared to **Re2**–**Re4** that results in a lower mass percentage of the former in the final products. These observations can be explained by substantial steric hindrance in the case of vinyl-containing phosphine (see Figure 1 and Figure 2), because in this case the distance between main polymer chain and bulky rhenium complex unit is shorter compared to that in the polymer based on styryl-containing phosphine that inevitably results in stronger repulsion between chain fragments in the former case.

The molecular weights (M_W_) of the Re-containing block copolymers estimated by GPC differs by more than 1 order of magnitude for high and low molecular RAFT agents (Table 1). For all copolymerization products a regular increase in the molecular weights and polydispersity was observed, due to the incorporation of the rhenium species.

A Dynamic Light Scattering (DLS) study of the copolymers shows that **p(VP-l/h-Re2)**–**p(VP-l/h-Re4)** spontaneously aggregates into nanoparticles upon dissolution in water with hydrodynamic diameters varying in the interval 180–285 nm (Table 1, Appendix A). The behavior of copolymers could be ascribed to their amphiphilic nature giving in aqueous solution the nanoparticles with hydrophobic rhenium core and hydrophilic polyvinylpyrrolidone corona. For **p(VP-l-Re1)** a histogram of size distribution (Appendix A) is broader to give a higher PDI, whereas in the case of **p(VP-h-Re1)** no particle formation was detected. The smaller content of the hydrophobic rhenium fragments relative to the hydrophilic part in the copolymers containing **Re1** obviously prevents the formation of stable nanoparticles, and in the case of **p(VP-h-Re1)**, where the hydrophilicity to hydrophobicity ratio is much higher, the compound can exist in water in the form of individual polymer chains similar to initial non-copolymerized p(VP).

### 2.2. Photophysical Properties

#### 2.2.1. Complexes **Re1**–**Re4**

The electronic absorption spectra of complexes **Re1**–**Re4** recorded in dilute methanol solution at room temperature (Appendix A) demonstrates two to three major transitions. The low-energy absorption bands located in the region between 340 and 500 nm could be assigned to metal-to-ligand charge transfer (^1^MLCT, dπ(Re) → π*(N^N)) transitions. More intense, higher-energy transitions are located at the wavelengths < 320 nm and are attributed to the spin-allowed π–π* ligand-centered (LC) excitations of the diimine skeleton and phosphine fragments, that leads to a noticeable difference in the position of these bands for the complex **Re1** compared to **Re2**–**Re4** containing diphenylvinylphosphine and 4-(diphenylphosphino)styrene, respectively. Both the energy and the molar absorption coefficients (Table 2) closely resemble those reported for analogous di-substituted Re(I) complexes containing phosphine as the ancillary ligand [21,22].

In methanol solution, the complexes under study display orange luminescence with structureless emission band profiles (Figure 3) and lifetimes in the microsecond domain, mainly characterized by the ^3^MLCT dπ(Re) → π*(N^N) nature of the excited states. The assignment of the phosphorescence to the ^3^MLCT character was made on the basis of obtained experimental results and theoretical analysis of closely analogous diphosphine Re(I) systems studied earlier [21,22]. The emission maxima for **Re1**–**Re4** are listed in Table 2, along with the respective emission quantum yields, excited state lifetime, and rate constants of radiative and nonradiative relaxation.

The emission bands of **Re1**–**Re4** demonstrate noticeable differences in the band maxima upon variations in the properties of diimine and phosphine ligands in the target coordination compounds. Namely, electron-donating substituents in the diimines induce moderate hypsochromic shifts of the emission bands due to the destabilization of their π* orbitals. In contrast, the introduction of the more electron-donating diphenylvinylphosphine, compared to 4-(diphenylphosphino)styrene, leads to the red-shift of the luminescence maximum observed for **Re1** that is attributed to the destabilization of the ground state d-orbitals localized on the rhenium metal center, and thus a reduction of the gap between the states taking part in emissive transition. Bulky substituents, both in the phosphine and diamine ligands, substantially increase emission quantum yield and lifetime due to blocking nonradiative (rotational and vibrational) relaxation channels, cf. the data (Table 2) for **Re1** vs. **Re2** (substitution of vinyl for styrene functions in the phosphine) and **Re2** vs. **Re3**, **Re4** vs. **Re3**.

#### 2.2.2. Copolymers **p(VP-l-Re)**, **p(VP-h-Re)**

Copolymers **p(VP-l/h-Re)** were thoroughly characterized with respect to their optical properties in methanol and aqueous solutions, Phosphate Buffered Saline (PBS, pH 7.4) solutions, as well as, for the selected compounds, in the model physiological media containing fetal bovine serum (FBS) and Dulbecco’s modified eagle medium (DMEM) via light absorption and photoluminescence spectroscopy (Figure 4 and Figure 5, Appendix A; Table 3). The UV-Vis absorption spectra of the metal-containing copolymers **p(VP-l/h-Re)** may be considered as a superposition of the spectral patterns from organic polymer (vinylpyrrolidone units) and rhenium complex components with the major contribution from the former (the high energy absorption in the 250–350 nm interval), which determines the general appearance of the spectral profile, shown in Figure 4. Nevertheless, one can also observe some spectral features (shoulders at ca. 300 and 400 nm) typical for the rhenium complexes absorption.

The copolymers described above are luminescent in the yellow (**p(VP-l/h-Re2)**–**p(VP-l/h-Re4)**) or orange (**p(VP-l/h-Re1)**) regions of the visible spectrum upon excitation at 365 nm which is another indication of the rhenium complexes incorporation into copolymer structure. A photoluminescence study of these copolymers revealed essential similarity of emission band profiles attributed to the Re(I) moiety with a slight hypsochromic shift of the band maxima compared to the starting rhenium complexes, cf. Table 2 and Table 3. The character of emission excited state can be evidently ascribed to ^3^MLCT (dπ(Re) → π*(N^N)) similar to the starting rhenium complexes and previously reported macromolecular rhenium(I) complexes [16,18]. Due to the lower emitter loading into **p(VP-l/h-Re1)**, its quantum efficiency is significantly reduced compared to the copolymers based on the Re(I) styrene-containing precursors (Table 3). It is also worth noting that the embedding of rhenium chromophores into polymeric matrix results in a slight but clearly visible increase in the phosphorescent lifetime obtained in methanol solutions (cf. Table 2 and Table 3). The effects (blue shift of emission maxima and longer lifetime) are substantially stronger upon the transfer of copolymer molecules into aqueous solution where they undergo self-assembly to give NPs, see Table 1. The electronic transitions in diimine rhenium(I) complexes associated with the MLCT states are known to be strongly affected by the rigidity of the environment (luminescent rigidochromism) [29,30,31], as well as the variations in the media polarity (solvatochromism) [32,33]. Thus, the observed hypsochromic shifts and increased emission lifetimes may be due to a less polar local environment of the chromophore and more rigid matrix, where the metal center is localized in the copolymer chains/NPs. 

These observations are in good agreement with the concept of nanoparticle formation (vide supra), where the hydrophobic rhenium core is protected from water by the hydrophilic polyvinylpyrrolidone corona. Less obvious variations in the emission characteristics in the case of **p(VP-l/h-Re1)** also correlate with the difference in the DLS data and indicate a lower trend to formation of NPs, and a less rigid structure of this copolymer in aqueous solution. An increase of **p(VP-l/h-Re)** emission lifetimes on going from aqueous solution to the phosphate buffered saline medium is observed due to the alteration of the ionic strength and, consequentially, compression of the electrical double layer of the copolymers, leading to the formation of more rigid copolymer conformations. It is also worth noting that for the copolymers containing the rhenium complexes with styrene functionality elongation of the polymer chain**, p(VP-l-Re#) → p(VP-h-Re#),** also gives higher lifetime values (both in aqueous and buffered solutions, see Table 3) evidently due to better chromophore protection from interaction with the solvent and additional rigidification of the Re environment as a consequence of the tighter packing of polymer chains into NPs.

In both aqueous and PBS solutions we found low sensitivity of emission parameters to molecular oxygen. The increase in emission lifetime values after deoxygenation reaches 33% and 15% for **p(VP-l/h-Re1)** and **p(VP-l/h-Re2)**-**p(VP-l/h-Re4)**, respectively (Table 3); the observed difference is presumably due to the better protection of the chromophoric centers by the polymeric styrene derivatives, compared to the vinyl congeners. One of the reasonable explanations of this effect can be also based on the difference in polymer structure in aqueous solutions for **p(VP-l/h-Re2)**–**p(VP-l/h-Re4)** and **p(VP-l/h-Re1)** due to the smaller number of rhenium centers in the latter, i.e., the smaller length of hydrophobic block, that leads to the less rigid environment and better oxygen penetration.

Among all copolymer samples, the most promising candidates for imaging applications are those loaded with **Re2** and **Re3** because they exhibit the highest quantum yields and emission lifetime. Nevertheless, we finally chose **p(VP-l-Re2)** and **p(VP-h-Re2)** copolymers because of the smallest difference in their emission lifetimes in aqueous media. Prior to starting cellular experiments, we studied the photophysical properties of the chosen copolymers in model physiological media (PBS, pH = 7.4 with addition of DMEM and FBS) (Figure 5, Table 3).

In general, photophysical behavior of **p(VP-l/h-Re2)** samples in model physiological media does not change significantly in comparison with other aqueous media. The excitation spectra monitored at the maxima of the dominant emission bands resembles the absorption spectra for the parent Re(I) species (Figure 5, dotted lines). The luminescence spectra of **p(VP-l/h-Re2)** upon excitation at 425 nm reveals the major yellow bands of the Re origin with minor contribution of the high energy component about 460 nm from the physiological media (Figure 5, solid lines). However, the feature to be noted is that the excited-state lifetimes before and after deoxygenation are essentially similar, indicating a lack of sensitivity of the **p(VP-l/h-Re2)** to the presence of molecular oxygen. This observation is indicative of the copolymer conformation, which completely protects the Re(I) emissive centers from quenching by molecular oxygen.

### 2.3. Biological Experiments

#### 2.3.1. Cell Experiments

The toxicity of the **Re2** complex and its conjugates with polymers was evaluated using the MTT method, see Figure 6. The toxicity of the copolymers was measured in relation to the molar concentration of the rhenium complex calculated from the values of the **Re2** loading into the copolymers, see Table 1. Comparative analysis of the data given in Figure 6 indicates that the complex is significantly more toxic even at relatively low (2.5 μM) concentrations, LC_50_ = 3.2 μM. Conjugation of the rhenium centers to the polymer reduces the copolymer toxicity and a viability rate above 80% is observed up to the concentration of 20 μM. Based on these findings, non-toxic concentrations of the copolymers, 5 μM, and moderately toxic concentration of **Re2**, 2.5 μM, were selected for the cell experiments under 24-h incubation.

Intracellular localization of the copolymers was investigated on living CHO-K1 cells, see Figure 7. Co-localization experiments with LysoTracker Deep Red (LTDR) display high Pearson’s and Manders’ coefficients that point to the copolymer’s localization in acidified cell compartments (endosomes and lysosomes). The co-localization with mitochondrial dye (BioTracker 405 Blue Mitochondria Dye, BTMD) is low, giving a Pearson’s correlation coefficient of about 18%. Manders’ overlap coefficient M2, showing the percentage of structures stained with the polymer that overlap with those stained with mitochondrial dye, is significantly higher than the M1 parameter. This observation can be explained by the fact that the mitochondrial dye signal occupies a large intracellular area, thus causing random unsystematic overlapping of the dye and sensor signals.

In order to compare the behavior of **Re2** and its copolymers in cells we also performed co-localization experiments for **Re2** on CHO-K1 cells. CHO-K1 cells were incubated for 1 h with the DMSO solution of **Re2** at two concentrations of 5 μM and 10 μM, and for 24 h with a 2.5 μM solution of the complex. The growing medium was then changed to fresh portion and the cells were stained with mitochondrial dye (15 min, 50 nM). Cells incubated with **Re2** for 24 h were additionally stained with lysosome-specific dye (30 min, 50 nM). It was found that the localization of the monomeric rhenium complex differs considerably from that of its copolymers. **Re2** readily internalizes into cells and spreads throughout the cytoplasmic region, providing a high degree of co-localization with mitochondrial dye. After 1 h of incubation with 5 μM solution of the complex, a pattern corresponding to normal distribution and morphology of mitochondria, e.g., many dense elongated structures (Figure 8, top row), was observed [34]. At higher concentrations and prolonged incubation, the formation of vesicular structures is observed (Figure 8, bottom row; Appendix A). Apparently, the addition of the complex causes a crucial change in mitochondria morphology to give so called mitochondria swelling; this phenomenon was reported for various toxic influences [35,36]. After 24 h of incubation, the complex is still distributed in vesicular structures that co-localize well with the mitochondrial dye (Appendix A). It also partially accumulates in structures that coincide in position with the lysotracker signal. From this, we can conclude that after 24 h of incubation, the process of metabolism of the dye by the cell takes place.

Thus, insertion of the complex into the structure of water-soluble copolymer significantly changed the biological properties of the final product, compared to the starting compound, to endow lower toxicity, higher solubility in aqueous solutions, and very different subcellular localization. The pathway of internalization, as well as localization for polymeric structures, significantly differs from the monomeric complex. Due to the balance of lipophilicity/hydrophilicity and small size, the complex very probably is able to cross the cell membrane by passive diffusion [37,38]. The positive charge of the complex evidently promotes its eventual localization in mitochondria observed in co-localization experiments. The copolymers containing the rhenium complex have a large molecular weight/size and are internalized into cells presumably by endocytic pathways, accumulating in acidified cellular compartments naturally related to this pathway of the probe uptake [39].

#### 2.3.2. PLIM Experiments

The lifetime of the excited state of the copolymers in the CHO-K1 cells, as well as the response of this parameter to deoxygenation, was investigated using phosphorescence lifetime imaging microscopy (PLIM). The sensitivity of the copolymers to oxygen was examined using the cells stained with **p(VP-l-Re2)** and **p(VP-h-Re2)** and maintained in a normal and nitrogen-saturated atmosphere (Figure 9). The lifetime of excited state decay was fitted by a two-exponential model. The average lifetime after deoxygenation changes by ca. 17% for **p(VP-l-Re2)** (from 1300 ns to 1540 ns) and ca. 9% for **p(VP-h-Re2)** (from 1100 ns to 1200 ns). The half-width of the lifetime distribution is ca. 500 ns for **p(VP-l-Re2)** and ca. 350 ns for **p(VP-h-Re2)**, that shows a percentage deviation of values from the average magnitude of ca. 15%. Thus, the variation of lifetime after deaeration is insignificant compared to the experimental uncertainty.

## 3. Materials and Methods


**General comments.**


All manipulations were carried out under anaerobic conditions. Starting materials from commercial sources were used as received, excluding *N*-vinylpyrrolidone and azobisisobutyronitrile which were vacuum-distillated and recrystallized from ethanol at 50 °C, respectively. [Re(N^N1)(CO)_3_(NCMe)](CF_3_SO_3_), [Re(N^N2)(CO)_3_(NCMe)](CF_3_SO_3_), and [Re(N^N3)(CO)_3_(NCMe)](CF_3_SO_3_) were prepared according to the conventional procedure [40]. Both RAFT 1 [26] and RAFT 2 [41] were synthesized according to the literature with a yield of ca. 80%. For the solution ^1^H, ^31^P{^1^H} NMR spectra were recorded on a Bruker Avance III 400 MHz spectrometer. Mass spectra were measured on a Bruker maXis II ESI-QTOF instrument in the ESI^+^ mode. IR spectra were recorded on IRAffinity-1S FTIR spectrophotometer (Shimadzu). Microanalyses were performed using a vario MICRO cube CHNS-analyzer (Elementar, Germany).

**X-ray Structure Determinations**. The crystals of **Re1** and **Re2** were immersed in cryo-oil, mounted in a Nylon loop, and measured at a temperature of 100 K. The diffraction data was collected with Agilent Technologies “Xcalibur” diffractometers using Cu Kα (λ = 1.54184 Å) radiation. A semi-empirical absorption correction (SADABS) [42] was applied to all data. The APEX2 [43] program package was used for cell refinements and data reductions. The structures were solved by direct methods using the SHELXT-2018 [44] program with the WinGX [45] graphical user interface. Structural refinements were carried out using SHELXL-2018 [46]. The crystallization solvent molecules in 1 could not be resolved unambiguously. The contribution of the missing solvent to the calculated structure factors was taken into account by using a SQUEEZE routine of PLATON [47]. The missing solvent was not taken into account in the unit cell content. All non-H atoms were anisotropically refined, and all hydrogen atoms were positioned geometrically and constrained to ride on their respective parent atoms with C–H = 0.95–1.00 Å and U_iso_ = 1.2–1.5U_eq_ (parent atom). The CF_3_SO_3_ counterion in **Re2** was disordered between two positions and refined with occupation factors of 0.63/0.37. Geometry and displacement constraints and restraints were applied to that motif. CCDC 2221681 and 2221684 contain the supplementary crystallographic data for this paper. These data are provided free of charge by the joint Cambridge Crystallographic Data Centre and Fachinformationszentrum Karlsruhe Access Structures service found at www.ccdc.cam.ac.uk/structures, accessed on 5 December 2022.


**Synthetic Procedure for Complexes [Re(CO)_2_(N**
**^N1–N**
**^N3)(P^0/1^)_2_]SO_3_CF_3_ (Re1–Re4).**


[Re(N^N)(CO)_3_(NCMe)](CF_3_SO_3_) (0.14 mmol), a stoichiometric amount of the corresponding phosphine ligands **P^0^** and **P^1^** (0.42 mmol), and butylated hydroxyanisole (5 wt%) were suspended in *o*-dichlorobenzene (2 mL) and degassed by purging nitrogen for 15 min with stirring. The reaction mixture was heated at 170 °C for 16 h in the sealing tube. The resulting dark orange (**Re2**–**Re4**) or red (**Re1**) solution was partially evaporated and purified using column chromatography (Silica gel 70–230 mesh, 1.5 × 20 cm, eluent dichloromethane/acetonitrile 10:1 *v*/*v* mixture). 


**[Re(CO)_2_(N**
**^N1)(P^0^)_2_]SO_3_CF_3_ (Re1).**


This was recrystallized by a gas-phase diffusion of diethyl ether into a chloroform solution of **Re1** at 4 °C to give an orangish crystalline material (59 mg, 43%). Crystals of complex **Re1** are block, C_40_H_34_N_2_O_2_P_2_ReS∙CF_3_O_3_S, *P* − 1, *a* = 11.9425(2), *b* = 14.14720(10), *c* = 14.26880(10) Å, *α* = 92.1510(10), *β* = 100.3300(10), *γ* = 98.0360(10), *V* = 2343.54(5) Å^3^, *Z* = 2, and *R_1_* = 0.0672, CCDC 2221681. IR (CH_2_Cl_2_; ν(CO), cm^−1^): 1942s, 1873s. ^1^H NMR (acetone-*d*_6_, 298 K; δ): 8.41 (d, *J*_HH_ = 5.6 Hz, 2H, 2,9-H bpy), 8.38 (d, *J*_HH_ = 8.3 Hz, 2H, 5,6-H bpy), 8.07 (t, *J*_HH_ = 7.7 Hz, 2H, 4,7-H bpy), 7.43−7.40 (m, 4H, *p*-H Ph), 7.35−7.31 (m, 10H, *m*-H Ph + 3,8-H bpy), 7.27−7.22 (m, *o*-H Ph 8H), 6.36−6.23 (m, 2H, P-CH = CH_2_), 5.98−5.86 (m, 2H, P-CH = CH_2_), and 5.36−5.27 (m, 2H, P-CH = CH_2_). ^31^P NMR (acetone-*d*_6_, 298 K; δ): 14.08. ESI^+^ MS (*m*/*z*): 823.17 [M]^+^ (calcd. 823.17). Analysis calculated for C_41_H_34_F_3_N_2_O_5_P_2_ReS: C, 50.67; H, 3.53; N, 2.88. Found: C, 50.73; H, 3.69; N, 2.77.


**[Re(CO)_2_(N**
**^N1)(P^1^)_2_]SO_3_CF_3_ (Re2).**


This was recrystallized by a gas-phase diffusion of diethyl ether into a chloroform solution of **Re2** at 4 °C to give a yellow crystalline material (146 mg, 93%). Crystals of complex **Re2** are prism, C_52_H_42_N_2_O_2_P_2_Re∙CF_3_O_3_S∙CHCl_3_∙CH_2_Cl_2_, *P* − 1, *a* = 13.9066(2), *b* = 14.7270(2), *c* = 15.2720(2) Å, *α* = 90.7680(10), *β* = 101.4770(10), *γ* = 116.004(2), *V* = 2736.85(8) Å^3^, *Z* = 2, and *R_1_* = 0.049, CCDC 2221684. IR (CH_2_Cl_2_; ν(CO), cm^−1^): 1939s, 1869s. ^1^H NMR (acetone-*d*_6_, 298 K; δ): 8.39 (d, *J*_HH_ = 8.2 Hz, 2H, 5,6-H bpy), 8.11 (d, *J*_HH_ = 5.6 Hz, 2H, 2,9-H bpy), 7.96 (t, *J*_HH_ = 7.7 Hz, 2H, 4,7-H bpy), 7.42−7.25 (m, 28H, Ph), 7.01 (t, *J*_HH_ = 6.5 Hz, 2H, 3,8-H bpy), 6.74 (dd, *J*_HH trans_ = 17.7, *J*_HH cis_ = 10.9 Hz, 2H, -Ph-CH = CH_2_), 5.88 (d, *J*_HH trans_ = 17.6 Hz, 2H, -Ph-CH = CH_2_), and 5.36 (d, *J*_HH cis_ = 11.0 Hz, 2H, -Ph-CH = CH_2_). ^31^P NMR (acetone-*d*_6_, 298 K; δ): 21.46. ESI^+^ MS (*m*/*z*): 975.23 [M]^+^ (calcd. 975.23). Analysis calculated for C_53_H_42_F_3_N_2_O_5_P_2_ReS: C, 56.63; H, 3.77; N, 2.49. Found: C, 56.45; H, 3.88; N, 2.38.


**[Re(CO)_2_(N**
**^N2)(P^1^)_2_]SO_3_CF_3_ (Re3).**


This was recrystallized by a gas-phase diffusion of diethyl ether into a dichloromethane solution of **Re3** at 4 °C to give a yellow crystalline material (142 mg, 82%). IR (CH_2_Cl_2_; ν(CO), cm^−1^): 1938s, 1868s. ^1^H NMR (acetone-*d*_6_, 298 K; δ): 8.45 (d, *J*_HH_ = 2.0 Hz, 2H, 5,6-H bpy), 7.81 (d, *J*_HH_ = 5.6 Hz, 2H, 2,9-H bpy), 7.41−7.24 (m, 28H, Ph), 6.96 (dd, *J*_HH_ = 6.0, 2.0 Hz, 2H, 3,8-H bpy), 6.73 (dd, *J*_HH trans_ = 17.7, *J*_HH cis_ = 11.0 Hz, 2H, -Ph-CH = CH_2_), 5.87 (d, *J*_HH trans_ = 17.7 Hz, 2H, -Ph-CH = CH_2_), 5.36 (d, *J*_HH cis_ = 11.0 Hz, 2H, -Ph-CH = CH_2_), and 1.35 (s, 18H, H ^t^Bu). ^31^P NMR (acetone-*d*_6_, 298 K; δ): 21.67. ESI^+^ MS (*m*/*z*): 1087.35 [M]^+^ (calcd. 1087.35). Analysis calculated for C_61_H_58_F_3_N_2_O_5_P_2_ReS: C, 59.26; H, 4.73; N, 2.27. Found: C, 59.10; H, 4.84; N, 2.19.


**[Re(CO)_2_(N**
**^N3)(P^1^)_2_]SO_3_CF_3_ (Re4).**


This was recrystallized by a gas-phase diffusion of diethyl ether into a chloroform solution of **Re4** at 4 °C to give a yellow crystalline material (118 mg, 72%). IR (CH_2_Cl_2_; ν(CO), cm^−1^): 1939s, 1868s. ^1^H NMR (acetone-*d*_6_, 298 K; δ): 8.48 (d, *J*_HH_ = 8.2 Hz, 2H, 4,7-H bpy), 8.06 (s, 2H, 5,6-H bpy), 7.59 (d, *J*_HH_ = 8.3 Hz, 2H, 3,8-H bpy), 7.36−7.32 (m, 4H, *p*-H Ph), 7.27−7.19 (m, 12H, *m*-H Ph), 7.06−6.97 (m, 12H, *p*-H Ph), 6.70 (dd, *J*_HH trans_ = 17.7, *J*_HH cis_ = 11.0 Hz, 2H, -Ph-CH = CH_2_), 5.85 (d, *J*_HH trans_ = 17.7 Hz, 2H, -Ph-CH = CH_2_), 5.35 (d, *J*_HH cis_ = 11.0 Hz, 2H, -Ph-CH = CH_2_), and 2.43 (s, 6H, CH_3_). ^31^P NMR (acetone-*d*_6_, 298 K; δ): 17.54. ESI^+^ MS (*m*/*z*): 1027.26 [M]^+^ (calcd. 1027.26). Analysis calculated for C_57_H_46_F_3_N_2_O_5_P_2_ReS: C, 58.21; H, 3.94; N, 2.38. Found: C, 57.94; H, 4.05; N, 2.40.

**Synthetic Procedure for the Polymers**. All compounds were synthesized with the reversible addition-fragmentation chain transfer polymerization (RAFT). Two types of the RAFT agents were used for the synthesis: S-benzyl O-ethylcarbonodithioate (RAFT agent 1) and S,S-dibenzyltrithiocarbonate (RAFT agent 2). Typically, N-vinylpyrrolidone (25 mmol for RAFT 1 and 47 mmol for RAFT 2), AIBN (10^−3^ M), and the RAFT agent (10^−2^ M) were diluted in dioxane (10 mL) and placed in a Pyrex reactor, degassed by three repeated freeze−evacuate−thaw cycles, and sealed. The polymerizations were carried out at 70 °C for 7 h for series 1 and 80 °C for 24 h for series 2. The obtained macro-RAFT agents were purified by re-precipitation and centrifugation from diethyl ether and distillated water. The yield equaled ca. 25% and 45% for each procedure, respectively. 

Low molecular p(VP) polymer; ^1^H NMR (400 MHz, D_2_O, δ): 7.58–7.40 (m, 0.1H), 4.12–3.67 (m, 1H), 3.67–3.28 (m, 2H), 2.74–2.38 (m, 2H), 2.38–2.12 (m, 2H), and 2.12–1.72 (m, 2H).

High molecular p(VP) polymer; ^1^H NMR (400 MHz, CD_2_Cl_2_, δ) 4.03–3.46 (m, 1H), 3.46–3.04 (m, 2H), 2.52–2.14 (m, 2H), 2.14–1.89 (m, 2H), and 1.77–1.36 (m, 2H).

**Synthetic Procedure for the copolymers p(VP-l-Re) and p(VP-h-Re).** The block copolymers were prepared by the copolymerization of the Re complexes (ca. 6 µmol) and macro-RAFT agents (100 mg) at 80 °C for 24 h. Both reagents were dissolved in dioxane (10 mL) and placed into the Pyrex reactor. AIBN (ca. 4 µmol) was then added, and the solution was degassed by three repeated freeze−evacuate−thaw cycles and sealed. The final block copolymers **p(VP-l/h-Re)** were re-precipitated three times with diethyl ether and dried. The solid obtained was dissolved in water, centrifugated to remove unreacted rhenium complexes, dialyzed (Orange Scientific; molecular weight cutoff = 3.5 kDa) for 3 days to remove the residual monomers and the solvent, and isolated from water by freeze-drying (FreeZone, Labconco). The yields were ca. 65 %.

**p(VP-l-Re1)**. ^1^H NMR (400 MHz, CD_2_Cl_2_, δ): 7.32–7.18 (m, 0.1H), 4.03–3.46 (m, 1H), 3.46–3.04 (m, 2H), 2.52–2.14 (m, 2H), 2.04–1.90 (m, 2H), and 1.90–1.36 (m, 2H); ^31^P NMR (400 MHz, CD_2_Cl_2_, δ): 13.96. 

**p(VP-l-Re2)**. IR (solid sample; ν(CO), cm^−1^): 1935w, 1860w. ^1^H NMR (400 MHz, CD_2_Cl_2_, δ): 7.32–7.18 (m, 0.1H), 4.03–3.46 (m, 1H), 3.46–3.04 (m, 2H), 2.52–2.14 (m, 2H), 2.04–1.90 (m, 2H), and 1.90–1.36 (m, 2H); ^31^P NMR (400 MHz, CD_2_Cl_2_, δ): 21.56. 

**p(VP-l-Re3)**. IR (solid sample; ν(CO), cm^−1^): 1935w, 1861w. ^1^H NMR (400 MHz, CD_2_Cl_2_, δ): 7.32–7.18 (m, 0.1H), 4.03–3.46 (m, 1H), 3.46–3.04 (m, 2H), 2.52–2.14 (m, 2H), 2.04–1.90 (m, 2H), and 1.90–1.36 (m, 2H); ^31^P NMR (400 MHz, CD_2_Cl_2_, δ): 21.52. 

**p(VP-l-Re4)**. IR (solid sample; ν(CO), cm^−1^): 1933w, 1859w. ^1^H NMR (400 MHz, CD_2_Cl_2_, δ): 7.32–7.18 (m, 0.1H), 4.03–3.46 (m, 1H), 3.46–3.04 (m, 2H), 2.52–2.14 (m, 2H), 2.04–1.90 (m, 2H), and 1.90–1.36 (m, 2H); ^31^P NMR (400 MHz, CD_2_Cl_2_, δ): 17.71. 

**p(VP-h-Re1)**. ^1^H NMR (400 MHz, CD_2_Cl_2_, δ): 4.03–3.46 (m, 1H), 3.46–3.04 (m, 2H), 2.52–2.16 (m, 2H), 2.10–1.90 (m, 2H), and 1.90–1.36 (m, 2H); ^31^P NMR (400 MHz, CD_2_Cl_2_, δ): 14.27. 

**p(VP-h-Re2)**. IR (solid sample; ν(CO), cm^−1^): 1935w, 1860w. ^1^H NMR (400 MHz, CD_2_Cl_2_, δ): 4.03–3.46 (m, 1H), 3.46–3.04 (m, 2H), 2.52–2.16 (m, 2H), 2.10–1.90 (m, 2H), and 1.90–1.36 (m, 2H); ^31^P NMR (400 MHz, CD_2_Cl_2_, δ): 21.48. 

**p(VP-h-Re3)**. IR (solid sample; ν(CO), cm^−1^): 1933w, 1858w. ^1^H NMR (400 MHz, CD_2_Cl_2_, δ): 4.03–3.46 (m, 1H), 3.46–3.04 (m, 2H), 2.52–2.16 (m, 2H), 2.10–1.90 (m, 2H), and 1.90–1.36 (m, 2H); ^31^P NMR (400 MHz, CD_2_Cl_2_, δ): 21.38.

**p(VP-h-Re4)**. IR (solid sample; ν(CO), cm^−1^): 1932w, 1857w. ^1^H NMR (400 MHz, CD_2_Cl_2_, δ): 4.03–3.46 (m, 1H), 3.46–3.04 (m, 2H), 2.52–2.16 (m, 2H), 2.10–1.90 (m, 2H), and 1.90–1.36 (m, 2H); ^31^P NMR (400 MHz, CD_2_Cl_2_, δ): 17.20. 

**NMR monitoring of copolymerization**. The ^1^H NMR spectroscopy was applied to determine the copolymerization of the rhenium complex to poly-*N*-vinylpyrrolidone. The experiment was performed using **Re2** and low molecular p(VP) polymer in the WILMAD ampoule overnight at 70 °C under the argon atmosphere on a Bruker Avance III 400 MHz spectrometer. The program multi_zgvd, with AU settings, was used with a 4-min interval for spectra recording. The recording was performed using a peak at 5.88 ppm of vinyl bound. The conversion was determined by the equation:(1)$$q=\frac{I_{0}-I_{n}}{I_{0}}$$
where q—conversion, *I*_0_—the integral intensity at the initial moment of the reaction, and *I_n_*—the integral intensity at a certain moment of the reaction.

**Gel Permeation Chromatography**. The molecular weights (M_W_) of (co)polymers were estimated by analytical gel permeation chromatography (GPC) on a LC-20AD chromatograph (Shimadzu) using the TSKgel SuperAW4000 column. The samples were dissolved in acetonitrile/0.1 M NaCl (20/80 vol.%). Chromatography was carried out at 30 °C with a sample aliquot of 20 µL and an eluent flow rate of 0.5 mL/min concerning the pullulan standards.

**Dynamic Light Scattering**. Particle sizes were determined by Dynamic Light Scattering using Malvern ZetaSizer NanoZS in water. The hydrodynamic diameter was taken as average of three individual measurements.

**Inductively Coupled Plasma Optical Emission Spectroscopy (ICPOES)**. The amount of Re complexes was determined by Inductively Coupled Plasma Optical Emission Spectroscopy (ICPE-9000, Shimadzu). The water-soluble Re(TPPTS)_2_ [10] was used as standard for calibration.

**Photophysical experiments**. All photophysical measurements in solution were carried out in dichloromethane. UV/Vis spectra were recorded using a Shimadzu UV-1800 spectrophotometer. The emission and excitation spectra in solution were measured with a Fluorolog-3 (HORIBA Jobin Yvon) spectrofluorimeter. The absolute emission quantum yield in solution was determined by a comparative method using [Ru(bpy)_3_][PF_6_]_2_ in aerated water (Φ_r_ = 0.042) [48] as the reference with the refraction coefficients of water and dichloromethane equal to 1.333 and 1.424, respectively (λ_exc_ = 365 nm).

To calculate the quantum yield, we used the following equation:(2)$$\Phi_{s}={\;\Phi }_{r}\frac{\eta_{s}^{2}A_{r}I_{s}}{\eta_{r}^{2}A_{s}I_{r}}$$
where Φ_s_ is the quantum yield of the sample, Φ_r_ is the quantum yield of the reference, *η* is the refractive index of the solvent, *A*_s_ and *A*_r_ are the absorbance of the sample and the reference at the wavelength of excitation, respectively, and *I*_s_ and *I*_r_ are the integrals of intensities of phosphorescence bands maxima. The lifetime of the complexes was determined by the time correlated single-photon counting (TCSPC) method combined with a Short-pulse laser DTL-375QT (355 nm, repetition frequency 1000 Hz), a Tektronix oscilloscope (DPO2012B, 100 MHz bandwidth), Ocean Optics scanning monochromator (Monoscan-2000, wavelength range 1 nm), FASTComTec digital time converter (MCS6 A1T4), and Hamamatsu counter head (H10682-01). The lifetime data were fitted using the Origin 9.0 software.

**Cell Culturing.** The Chinese hamster ovary CHO-K1 cells were maintained in DMEM/F12 (Biolot, St. Petersburg, Russia) medium supplemented with 10% FBS (Gibco, Carlsbad, CA, USA), 2 mM glutamine (Gibco, Carlsbad, CA, USA), and penicillin/streptomycin at a concentration of 100 U/mL (Thermo Fisher Scientific, Waltham, MA, USA). The cell culture was maintained in a humidified incubator at 37 °C with 5% CO_2_ and passaged routinely using trypsin-EDTA (Thermo Fisher Scientific, Waltham, MA, USA). For live-cell confocal microscopy, the cells, at a concentration of 1 × 10^5^ CHO-K1 cells per 1 mL of growing media, were seeded in glass bottom 35 mm dishes (Ibidi GmbH, Gräfelfing, Germany) and incubated for 48 h until they reached a confluence of ~70%. Complex **Re2** was dissolved in DMSO to a concentration of 5 mM, mixed with growing media, and added to the cells to a final concentration of 2.5, 5 and 10 µM (content of DMSO in the growing media did not exceed 0.2%). The copolymers were dissolved in water, filtered through 0.20 µm PVDF syringe filter, and added to the cells to a final concentration of 5 µM (concentration of **Re2** according to the content of the complex in copolymers). After incubation with the probe for 24 h, cells were washed with fresh media with all supplements.

**MTT Assay.** CHO-K1 cells were seeded in 96-well plates (Nunc, Thermo Fisher Scientific, Waltham, MA, USA) using 1 × 10^4^ cells in 100 µL of culture medium/well and incubated overnight. The complex was dissolved in DMSO at a concentration of 5 mM and diluted with growing media to a concentration of 1–20 µM (content of DMSO in the growing media did not exceed 0.4%). The polymers were dissolved in water and added to the cells to a final Re concentration of 1–20 µM. After incubation for 24 h, the cells were treated with the MTT reagent 3(4,5-dimethyl-2-thiasolyl)-2,5-diphenyl-2H-tetrasole bromide (Thermo Fisher Scientific, Waltham, MA, USA) at the concentration of 0.5 mg/mL. After further incubation at 37 °C under 5% CO_2_ for 2.5 h, the media was removed, and the formazan crystals were dissolved in DMSO (Merck, Munich, Germany). The absorbance in each well was measured at 570 nm using a SPECTROstar Nano microplate reader (BMG LABTECH, Ortenberg, Germany). Viability was determined as a ratio of the average absorbance value of the wells containing conjugate to that of the control. Six repetitions were performed for each concentration of the conjugates. The results are shown as mean ± standard deviation.

**Lysosomal and mitochondrial staining.** For the vital staining of mitochondria in CHO-cells, BioTracker 405 Blue Mitochondria Dye (Sigma Aldrich, Merck, Munich, Germany) was used at the concentration of 50 nM. LysoTracker Deep Red (Thermo Fisher Scientific, Waltham, MA, USA) was used at the concentration of 50 nM for the vital staining of acidified late endosomes and lysosomes. Incubation with the complex (5 µM and 10 µM, 1 h; 2.5 µM, 24 h) and polymers (5 µM, 24 h) was followed by rinsing the cells with fresh media 3 × 1 mL and incubation with a new portion of growing media for 15 min. BioTracker and LysoTracker were added to the cells for 15 and 30 min, respectively, prior to confocal imaging. The LC_50_ value was calculated using the AAT Bioquest tool [49].

**Confocal Microscopy.** The imaging of living CHO-K1 cells was carried out by using a confocal inverted Nikon Eclipse Ti2 microscope (Nikon Corporation, Tokyo, Japan) with a 40× water immersion objective. The required temperature, 5% CO_2_ and required oxygen percentage during the experiments were maintained using a Stage Top Incubator Tokai HIT (Japan) equipped with Digital gas mixer GM-8000. The emission of the probes was excited at 405 nm, and the signal was registered in the red (570–620 nm) channel. The fluorescence of BioTracker 405 Blue Mitochondria Dye was excited at 405 nm and recorded at 425–475 nm. LysoTracker Deep Red fluorescence was excited at 637 nm and recorded at 663–738 nm. Differential interference contrast (DIC) images complemented each luminescent confocal microphotograph. The images were processed and analyzed using ImageJ software (National Institutes of Health, Bethesda, MY, USA). Quantitative co-localization analysis and the determination of Pearson (P) and Manders’ (M1 and M2) co-localization coefficients were carried out using ImageJ JACoP Plugin. Thresholds for M1 and M2 calculations were set by a visually estimated value for each channel. The results are presented as mean ± standard deviation calculated for 4–5 microphotographs.

**PLIM**. Phosphorescence Lifetime Imaging Microscopy (PLIM) of CHO-K1 cells was carried out using a time-correlated single-photon counting (TCSPC) DCS-120 module (Becker&Hickl GmbH, Berlin, Germany) integrated into the Nikon Eclipse Ti2 confocal instrument. Emission was excited with a picosecond laser at 405 nm, phosphorescence was recorded using a 575 nm long pass filter, and a 630/75 nm band pass filter with a pinhole of 0.5–1.0. The following settings were used: frame time 20.51 s, pixel dwell time 77.70 µs, points number 1024, time per point 75.00 ns, time range of PLIM recording 76.80 µs, total acquisition time 120–140 s, and image size 512 × 512 pixels. Water immersion 40× objective with zoom 5.33 provided a scan area of ca. 0.08 mm × 0.08 mm. Phosphorescence lifetime data were processed with SPCImage 8.1 software (Becker & Hickl GmbH, Berlin, Germany) using biexponential decay modes with an average goodness of the fit 0.8 ≤ χ^2^ ≤ 1.2. The average number of photons per curve was not less than 2500 at binning 8–9. The colors in the PLIM images show intensity-weighted average lifetime calculated for a two-exponential decay fit, τ_i_ = (A_1_τ_1_^2^ + A_2_τ_2_^2^) / (A_1_τ_1_ + A_2_τ_2_), where A_i_ is the weight of the i-exponent and t_i_ is the corresponding lifetime component.

## 4. Conclusions

Herein, a novel and successful strategy to achieve the decent water-solubility and biocompatibility for the rhenium(I) complexes via copolymerization reaction is presented. The approach consists in the preparation of Re(I) precursors, **Re1**–**Re4**, containing the phosphine ligands with polymerizable vinyl and styryl functions, and further utilization of the obtained precursors in copolymerization with the N-vinylpyrrolidone-based water-soluble Macro-RAFT agents. The copolymerization reactions gave two families of block copolymers, **p(VP-l-Re#)** and **p(VP-h-Re#)**, containing the rhenium chromophores and differing in the polymer chain length. Dynamic Light Scattering investigation of the copolymers indicates the spontaneous aggregation of **p(VP-l/h-Re2)**–**p(VP-l/h-Re4)** into nanoparticles in aqueous solution, while in the case of **p(VP-l-Re1)** no particle formation was detected. All complexes and block copolymers **p(VP-l/h-Re#)** luminesce in fluid media from the ^3^MLCT (dπ(Re) → π*(N^N)) excited state with lifetimes in the microsecond domain. In aqueous and PBS solutions, the copolymers display a clearly visible trend which consist in the blue shift of emission maxima and higher lifetime upon the polymeric emitter transfer from methanol to water, that is related to the copolymers self-assembly into NPs to give less polar and more rigid local environment of the chromophore. The elongation of the starting polymer chain, as well as increase in salinity of the water solution, also gives longer excited state lifetime due to better chromophore protection from interaction with the solvent in the former case and compression of the NPs corona in the latter. The **p(VP-l-Re2)** and **p(VP-h-Re2)** copolymers were tested in biological experiments using CHO-K1 cell line along with the initial **Re2** complex. The MTT assay showed that conjugation of the Re(I) moiety with the polymer chains considerably reduced the toxicity of the final Re-containing polymeric species. The copolymers mainly display localization in endosomes, whereas the **Re2** complex shows localization and accumulation in mitochondria, leading to mitochondrial swelling. PLIM experiments on the CHO-K1 cells revealed that excited state lifetimes of **p(VP-l-Re2)** and **p(VP-h-Re2)** in aerated and nitrogen-saturated atmosphere are nearly identical, that is very similar to the chromophores behavior in solution and indicates effective shielding of the chromophore from interaction with molecular oxygen in the polymeric species. 

## Data Availability

Data is contained within the article and Supplementary Material.

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
