# Peer review of "Rhenium(I) Block Copolymers Based on Polyvinylpyrrolidone: A Successful Strategy to Water-Solubility and Biocompatibility"

_molecules, 2023, doi:10.3390/molecules28010348_

Round 1
Reviewer 1 Report
This paper deals with Re-complex contanining block-copolymers for biosenseor using their potophysical properties.
The quality of manuscript, signficance and acheivement of this study must be worth publishing in Molecules almost as it is.
Before acceptance, however, some points should be explained to add some sentences in the text.
(1) The merit of water-soluble block-copolymers rather than direct use of [Re(CO)2L2]n+ complex.
(2) Stability (keeping CO) of [Re(CO)2L2]n+ against photo-irradiation.
(3) Block-copolymer was synthesised uniformly.
In addition, some minor points should be corrected.
(1) L206: It worth noting -> It is worth noting
(2) L222: Inductively -> inductively or
Inductively coupled plasma -> Inductively Coupled Plasma
That's all.
Author Response
Please, see the attachment

Reviewer 2 Report
Journal: Molecules
Manuscript Number: Molecules-2124933
Title: Rhenium(I) block-copolymers based on Polyvinylpyrrolidone: successful strategy to water-solubility and biocompatibility
In this paper, the authors designed a series of diphosphine Re(I) complexes Re1−Re4. These complexes were copolymerized with N-vinylpyrrolidone-based Macro-RAFT agents of different polymer chain lengths to give water-soluble copolymers of low-molecular p(VP-l-Re) and high-molecular p(VP-h-Re) block-copolymers containing rhenium complexes. I have carefully examined the manuscript. I recommend the manuscript for publication in Journal of Molecules after modification. My comments are as follows.
Comments:
(1) There were some issues in the figures. In the Scheme 2, the reaction condition was labeled incorrectly in the synthesis route using RAFT2, in which “a)” was labeled as “b)”. In the Figures 2-5, the decimal points of the vertical coordinate were incorrectly written as the commas. Please check the manuscript and revise these problems.
(2) The data of Figure 4 had not been analyzed in the manuscript, please add the relevant description.
(3) In Page 3 Lines 116-118, Page 6 Lines 213-215 and the Section 3, the data of IR spectra were mentioned, but IR spectra was not shown in the manuscript or Electronic Supporting Information. Suggest the authors add it.
(4) “These observations can be explained by substantial steric hindrance…” was mentioned in Page 6 Lines 229-231. Please explain in detail how steric hindrance affects the reactivity of Re in combination with specific molecular structures.
(5) In Page 6 Lines 234-235, the authors mentioned that “…a regular increase in the molecular weights and polydispersity was observed…”. Please explain the cause of this phenomenon.
(6) “…which determines the general appearance of the spectral profile, Figure 3” was mentioned in Page 8 Lines 298-302. The Figure 3 only showed the comparison of normalized excitation and emission spectra of Re1-Re4 complexes. How to draw this conclusion from this figure, please explain it.
(7) In the Conclusions, it is recommended that the authors streamline the conclusions of this manuscript to facilitate readers to better extract information.
(8) As for references, this manuscript had few references from the last five years. The new references must be added to show the latest progress of related research and prove the meaning of this work. In addition, the format of reference should be modified according to the requirement.
Author Response
Please, see the attachment

Reviewer 3 Report
Dear Editor:
This is my comments about the article titled " Rhenium(I) block-copolymers based on Polyvinylpyrrolidone: 2 successful strategy to water-solubility and biocompatibility ". This article was interested in
A series of diphosphine Re(I) complexes Re1−Re4 has been designed via decoration of 12the archetypal core {Re(CO)2(N^N)} through the installations of the phosphines P0 and P1 bearing 13the terminal double bond, where N^N = 2,2′-bipyridine (N^N1), 4,4'-di-tert-butyl-2,2'-bipyridine 14(N^N2) or 2,9-dimethyl-1,10-phenanthroline (N^N3) and P0 = diphenylvinylphosphine, P1 = 154-(diphenylphosphino)styrene. This article is in good production and the work at all is interesting and could be accepted for publication after a minor revision.
My comments
1- The aim of the work needs modification
2- I asked the authors to add these references in the introduction part to strength the background
- Arabian Journal for Science and Engineering, 1-18, 2022
-Journal of Molecular Liquids 351, 118620, 2022
- Journal of Molecular Liquids 345, 117803, 2022
3- The discussion parts are highly detailed, the authors must focused on essential data only and reduced all parts
4-Conclusion must be rewritten in more clear form.
5-Finally the authors must revise language of the manuscript before publication and the whole article must be adjusted based on journal style.
Author Response
Please, see the attachment
